# The Use of COVID-19 Mobile Apps in Connecting Patients with Primary Healthcare in 30 Countries: Eurodata Study

**DOI:** 10.3390/healthcare12141420

**Published:** 2024-07-16

**Authors:** Raquel Gómez-Bravo, Sara Ares-Blanco, Ileana Gefaell Larrondo, Lourdes Ramos Del Rio, Limor Adler, Radost Assenova, Maria Bakola, Sabine Bayen, Elena Brutskaya-Stempkovskaya, Iliana-Carmen Busneag, Asja Ćosić Divjak, Maryher Delphin Peña, Philippe-Richard Domeyer, Sabine Feldmane, Louise Fitzgerald, Dragan Gjorgjievski, Mila Gómez-Johansson, Miroslav Hanževački, Oksana Ilkov, Shushman Ivanna, Marijana Jandrić-Kočić, Vasilis Trifon Karathanos, Erva Ücüncü, Aleksandar Kirkovski, Snežana Knežević, Büsra Çimen Korkmaz, Milena Kostić, Anna Krztoń-Królewiecka, Liga Kozlovska, Heidrun Lingner, Liubovė Murauskienė, Katarzyna Nessler, Naldy Parodi López, Ábel Perjés, Davorina Petek, Ferdinando Petrazzuoli, Goranka Petricek, Martin Sattler, Bohumil Seifert, Alice Serafini, Theresa Sentker, Gunta Ticmane, Paula Tiili, Péter Torzsa, Kirsi Valtonen, Bert Vaes, Shlomo Vinker, Ana Luisa Neves, Marina Guisado-Clavero, María Pilar Astier-Peña, Kathryn Hoffmann

**Affiliations:** 1Centre Hospitalier Neuro-Psychiatrique, CHNP, 43, Avenue des Alliés, L-9012 Ettelbruck, Luxembourg; 2Research Group Self-Regulation and Health, Institute for Health and Behaviour, Department of Behavioural and Cognitive Sciences, University of Luxembourg, Campus Belval, Maison des Sciences Humaines 11, Porte des Sciences, L-4366 Esch-sur-Alzette, Luxembourg; 3Federica Montseny Health Centre, Gerencia Asistencial de Atención Primaria, Servicio Madrileño de Salud, Avenida de la Albufera, 285, 28038 Madrid, Spain; 4Medical Specialties and Public Health, School of Health Sciences, University Rey Juan Carlos, Avda. de Atenas, s/n., 28922 Alcorcón, Spain; 5Instituto de Investigación Sanitaria Gregorio Marañón, calle del Dr. Esquerdo, 46, 28007 Madrid, Spain; 6Fundación de Investigación e Innovación Biosanitaria de Atención Primaria (FIIBAP), 28003 Madrid, Spain; 7Research Network on Chronicity, Primary Care and Health Promotion-RICAPPS-(RICORS), 28029 Madrid, Spain; 8Department of Family Medicine, Faculty of Medicine, Tel Aviv University, Tel Aviv 6997801, Israelvinker01@gmail.com (S.V.); 9Department Urology and General Practice, Faculty of Medicine, Medical University of Plovdiv, 4000 Plovdiv, Bulgaria; 10Research Unit for General Medicine and Primary Health Care, Faculty of Medicine, School of Health Science, University of Ioannina, 451 10 Ioannina, Greece; 11Department of General Practice, Faculté de Médicine Henri Warembourg, University of Lille, 59045 Lille, CEDEX 1, France; 12General Medicine Department, Belarusian State Medical University, 220116 Minsk, Belarus; lenabs@tut.by; 13Kinetic Therapy and Special Motricity, Faculty of Physical Education and Sport, “Spiru Haret” University, 030045 Bucharest, Romania; 14Health Centre Zagreb Centar, 10000 Zagreb, Croatia; 15Department of Geriatric Medicine, Hôpitaux Robert Schuman, L-1130 Luxembourg, Luxembourg; 16School of Social Sciences, Hellenic Open University, 263 35 Patra, Greece; 17Department of Family Medicine, Faculty of Medicine, Rīga Stradins University, LV-1007 Riga, Latvia; 18Irish College of General Practice (MICGP), Royal College of Physician (MRCSI), D02 YN77 Dublin, Ireland; 19Center for Family Medicine, Medical Faculty Skopje, 1000 Skopje, North Macedonia; 20Capio Kvillebäcken Health Centre, 417 22 Gothenburg, Sweden; 21Department of Family Medicine, “Andrija Stampar” School of Public Health, School of Medicine, University of Zagreb, 10000 Zagreb, Croatia; 22Health Centre Zagreb West, 10000 Zagreb, Croatia; 23Department of Family Medicine and Outpatient Care, Medical Faculty, Uzhhorod National University, Narodna Square, 3, 88000 Uzhhorod, Transcarpathian Region, Ukraineivshushman@gmail.com (S.I.); 24Health Center Krupa Na Uni, 79227 Krupa na Uni, Bosnia and Herzegovina; 25Medical Education Unit, Laboratory of Hygiene and Epidemiology, Medical Department, Faculty of Health Sciences, University of Ioannina, 451 10 Ioannina, Greece; 26General Health System (GHS) Cyprus, 6037 Larnaca, Cyprus; 27Department of Family Medicine, Prof. Dr. Cemil Tascioglu City Hospital, 34384 Istanbul, Turkey; 28Faculty of Medicine, Ss. Cyril and Methodius University, 1000 Skopje, North Macedonia; 29Department of Medical Sciences, Academy of Applied Studies Polytechnic, 11000 Belgrade, Serbia; lesta59@yahoo.com; 30Van Gürpınar District Public Hospital, 65900 Gürpınar, Turkey; 31Health Center “Dr. Đorđe Kovačević”, 11550 Lazarevac, Serbia; 32Department of Family Medicine, Andrzej Frycz Modrzewski Krakow University, 30-705 Krakow, Poland; 33Department of Family Medicine, Riga Stradins University, LV-1007 Riga, Latviagunta.t@inbox.lv (G.T.); 34Rural Family Doctors’ Association of Latvia, LV-4501 Balvi, Latvia; 35Center for Public Health and Healthcare, Department of Medical Psychologie OE5430, Hannover Medical School, 30625 Hannover, Germany; 36Department of Public Health, Institute of Health Sciences, Faculty of Medicine, Vilnius University, 03101 Vilnius, Lithuania; liubove.murauskiene@mf.vu.lt; 37Department of Family Medicine, Uniwersytet Jagielloński—Collegium Medicum (UJCM), 31-061 Krakow, Poland; 38Department of Pharmacology, Sahlgrenska Academy, University of Gothenburg, 405 30 Gothenburg, Sweden; 39Department of Family Medicine, Semmelweis University, 1085 Budapest, Hungaryptorzsa@gmail.com (P.T.); 40Department of Family Medicine, Faculty of Medicine, University of Ljubljana, Vrazov trg 2, 1000 Ljubljana, Slovenia; davorina.petek@gmail.com; 41Department of Clinical Sciences in Malmö, Centre for Primary Health Care Research, Lund University, 221 00 Malmö, Sweden; ferdinando.petrazzuoli@gmail.com; 42European Parliament, L-2929 Luxembourg, Luxembourg; 43Institute of General Practice, First Faculty of Medicine, Charles University, Albertov 7, 110 00 Prague, Czech Republic; 44Azienda Unità Sanitaria Locale di Modena, Laboratorio EduCare, University of Modena and Reggio Emilia, 41121 Reggio Emilia, Italy; 45Communicable Diseases and Infection Control Unit, Wellbeing Services, County of Vantaa and Kerava, P.O. Box 341, 01301 Vantaa, Finland; 46Faculty of Medicine, University of Helsinki, P.O. Box 63, 00014 Helsinki, Finland; 47Department of Public Health and Primary Care, KU Leuven, 3000 Leuven, Belgium; bert.vaes@kuleuven.be; 48Department of Primary Care and Public Health, Imperial College London, London SW7 2AZ, UK; ana.luisa.neves14@imperial.ac.uk; 49Investigation Support Multidisciplinary Unit for Primary Care and Community North Area of Madrid, 28035 Madrid, Spain; marina.guisado@gmail.com; 50Universitas Health Centre, SALUD (Servicio Aragonés de Salud), University of Zaragoza, Andres Vicente 42, 50009 Zaragoza, Spain; 51Department of Primary Care Medicine, Medical University of Vienna, 1090 Vienna, Austria

**Keywords:** app, COVID-19, e-health, health information interoperability, primary healthcare

## Abstract

Background: The COVID-19 pandemic has necessitated changes in European healthcare systems, with a significant proportion of COVID-19 cases being managed on an outpatient basis in primary healthcare (PHC). To alleviate the burden on healthcare facilities, many European countries developed contact-tracing apps and symptom checkers to identify potential cases. As the pandemic evolved, the European Union introduced the Digital COVID-19 Certificate for travel, which relies on vaccination, recent recovery, or negative test results. However, the integration between these apps and PHC has not been thoroughly explored in Europe. Objective: To describe if governmental COVID-19 apps allowed COVID-19 patients to connect with PHC through their apps in Europe and to examine how the Digital COVID-19 Certificate was obtained. Methodology: Design and setting: Retrospective descriptive study in PHC in 30 European countries. An ad hoc, semi-structured questionnaire was developed to collect country-specific data on primary healthcare activity during the COVID-19 pandemic and the use of information technology tools to support medical care from 15 March 2020 to 31 August 2021. Key informants belong to the WONCA Europe network (World Organization of Family Doctors). The data were collected from relevant and reliable official sources, such as governmental websites and guidelines. Main outcome measures: Patient’s first contact with health system, governmental COVID-19 app (name and function), Digital COVID-19 Certification, COVID-19 app connection with PHC. Results: Primary care was the first point of care for suspected COVID-19 patients in 28 countries, and 24 countries developed apps to complement classical medical care. The most frequently developed app was for tracing COVID-19 cases (24 countries), followed by the Digital COVID-19 Certificate app (17 countries). Bulgaria, Italy, Serbia, North Macedonia, and Romania had interoperability between PHC and COVID-19 apps, and Poland and Romania’s apps considered social needs. Conclusions: COVID-19 apps were widely created during the first pandemic year. Contact tracing was the most frequent function found in the registered apps. Connection with PHC was scarcely developed. In future pandemics, connections between health system levels should be guaranteed to develop and implement effective strategies for managing diseases.

## 1. Introduction

The COVID-19 pandemic has changed the organization of the health systems across Europe to provide medical care for COVID-19 patients [1]. Numerous primary care facilities have adopted remote consultations as a preferred method while reserving in-person appointments for patients with more severe or acute symptoms [2]. Technological solutions such as contact tracing applications (apps) and COVID-19 symptom checkers have become widely used tools to identify possible cases. However, their connection with the initial point of access in the healthcare system remains unclear. Various mobile apps have been developed by countries, private companies, and other entities to assist in healthcare [3,4]. While the use of apps in general has become widespread and their potential in detecting COVID-19 cases is promising, there is limited evidence available regarding the benefits of e-health, such as m-health and teleconsultation. Further research is necessary to enhance future interventions [5]. Previous studies have identified inequalities in the use of e-health, highlighting the need for measures to ensure universal and equitable access [6].

In Europe, many countries launched mobile contact-tracing apps, in line with the guidance from the European Centre for Disease Prevention and Control guidelines (ECDC) [7]. Functionalities varied across countries; some apps allowed for the exchange of information with apps from other countries [8]. However, the COVID-19 contact-tracing apps have not been widely adopted by the European populations [4,8,9].

Challenges remain in terms of expanding the use of digital tools and evaluating clinical care in the context of COVID-19 [8,10]. In July 2021, the European Union (EU) implemented the use of the EU Digital COVID-19 Certificate to travel from one state to another within the EU and affiliated countries (Appendix A). This required demonstrating immunity to COVID-19 by vaccination, being recently cured of the disease, or having a negative COVID-19 test [11]. Countries have developed applications to issue the EU COVID-19 digital certificate almost automatically, thus reducing bureaucracy. The majority of COVID-19 cases have been treated in the community, and contact tracing was performed between public health and primary care depending on the country [4], but the connection among COVID-19 apps and primary healthcare (PHC) has not been described. The main aim of this study is to examine if governmental COVID-19 apps allowed COVID-19 patients to connect with PHC through their apps in Europe. The secondary aim was to examine how the Digital COVID-19 Certificate was obtained in mobile apps in Europe.

## 2. Methods

### 2.1. Design of the Study

This study was retrospective and descriptive, and it was conducted starting in 2022 collecting information from 15 March 2020 to 31 August 2021 in 30 European countries (Figure 1).

### 2.2. Participants

Key informants who were linked to the working group (EGPRN) and the World Organization of Family Doctors (WONCA) in Europe. These researchers belonged to 30 European countries (Austria, Belarus, Belgium, Bosnia and Herzegovina, Bulgaria, Croatia, Czech Republic, Cyprus, Finland, France, Germany, Greece, Hungary, Ireland, Israel, Italy, Latvia, Lithuania, Luxembourg, North Macedonia, Poland, Portugal, Romania, Serbia, Slovenia, Spain, Sweden, Turkey, Ukraine, and the United Kingdom). Researchers received an open invitation through both networks to participate. We extended invitations to 80 contacts within these networks to participate. Of these, 46 general practitioners (GPs) agreed to serve as researchers, along with one public health expert who maintains close ties with local GPs, and one medical student collaborating with a participating GP. Among the participants, 42 GPs were actively practicing during the pandemic, while 35 GPs were affiliated with university departments.

### 2.3. Variables

The main variables examined in this study were the patient’s initial contact with the healthcare system, the existence of COVID-19 hotlines, the type of COVID-19 mobile app used, and the connection between the COVID-19 mobile app and PHC. The connection between the COVID-19 mobile app and PHC was defined as the ability to share information from the COVID-19 app with one’s PHC provider through means such as phone call, email, mobile messaging service, online appointments, or other relevant technologies in the case of suspected COVID-19. Please refer to Appendix A for detailed descriptions of these terms.

### 2.4. Data Collection

An ad hoc, semi-structured questionnaire intended to provide country-specific data on PHC activity in the COVID-19 pandemic was developed. The questionnaire also collected information on the use of information technology (IT) tools to support medical care (Table 1). To create the initial questionnaire, we first reviewed the ECDC guidelines on developing contact-tracing apps.

Subsequently, the European Commission curated a list of contact-tracing and COVID-19 certificate apps from EU countries on a webpage. The core group (4 GPs, one of whom is also a public health doctor) then assessed the English-language information for these national websites, analyzing their types, basic data collection, and functionalities. Based on this analysis, a second version of the questionnaire was crafted. After collecting this information, it was shared with all key informants for their input, feedback, or suggestions to enhance the questionnaire. Two online meetings were organized to share comments until a consensus on the questionnaire items was reached. Following a month-long period, the final version of the questionnaire was completed. The national key informants received recommendations to collect information from relevant and reliable official sources (governmental websites and governmental guidelines are available in Appendix A). One or two national key informants per country filled out the questionnaire; it was peer-reviewed by a different national researcher before sending it to the core group. They checked the national data to assure the data quality.

### 2.5. Analysis

Qualitative variables from the national questionnaires were organized and summarized. The core group performed an international peer review of all the national data collected. In the case of disagreement, the core group contacted the national key informants to clarify the description provided. Language was standardized for comparisons under the advice of all the key informants using English. All national key informants reviewed the results to confirm the findings. The STROBE guidelines of this study can be checked in Appendix A.

## 3. Results

In 28 countries, primary care served as the initial point of contact for suspected COVID-19 patients, often in collaboration with other resources. Additionally, 25 countries established COVID-19 hotlines to provide information and address concerns related to the virus. Contact-tracing mobile apps were adopted by 24 countries (as shown in Table 2 and Figure 2). All countries developed additional functionalities tailored to their contact-tracing apps. The most commonly featured functionality, found in at least 12 countries, was providing health advice related to COVID-19, followed by a self-assessment of symptoms, available in 11 countries.

Turkey’s app sent notifications to local authorities in the case of non-compliance with isolation measures. Five countries (Bulgaria, Italy, North Macedonia, Serbia, and Romania) integrated COVID-19 mobile app information with PHC systems for each patient. In Bulgaria, GPs received an email notification if patients were identified as having a high risk for COVID-19 after a self-assessment test. GPs then decided how to proceed, either through a phone call or further examination. GPs could access the app using an e-signature to review patient results. In Italy (specifically the Campania region), the e-COVID app allowed patients to submit reports to their GPs. The Tests and Swabs app facilitated proactive care from GPs for COVID-19 patients in the Campania region. In North Macedonia, patient symptoms and COVID-19 test results were directly downloaded into an online portal and app called Moj Termin. This app was already in use by GPs before the pandemic, and GPs were required to fill out daily forms for their patients regarding COVID-19 symptoms. In Romania, patients could directly send reports to their GPs through the app. In Serbia, patients could complete a self-assessment test, and if the results indicated potential serious COVID-19 symptoms, they were recommended to schedule an examination with a doctor. Serbian citizens could use the national app to contact GPs or ask questions, which would then appear in the electronic health records of Serbian GPs, enabling direct communication with patients. Poland and Romania had apps that considered the social needs of the population. In Poland, isolated patients could contact social workers for support through the Polish app. The Romanian app was designed to assist Romanians living abroad in connecting with people when seeking help for their social support and translation service needs.

In Slovenia, the apps for COVID-19 testing and COVID-19 vaccination were able to transfer information to the electronic health record (EHR). However, there was no connection established between the clinical data from these apps and the EHR.

In Finland, there was an existing web portal called Omaolo that offered various functionalities such as sharing symptoms, booking PHC appointments, and communicating with healthcare providers. This portal was updated to include COVID-19 features for contacting PHC, but the specific functionalities varied across municipalities and regions. In June 2024, six apps were active.

Additionally, apps were developed for the issuance of the Digital COVID-19 Certificate. Portugal, Spain and the United Kingdom used patient medical record to obtain the Digital COVID-19 Certificate. In Austria, citizens were required to independently upload their vaccine and recovery certificates into the official national Green Pass app. These certificates could be obtained from physicians, pharmacies, or accessed through the electronic health portal ELGA. In the Czech Republic, the online patient portal transferred COVID-19 vaccination information to an app, but without the functionalities of the portal. In France, Ireland, and Serbia, contact-tracing apps were utilized to obtain the Digital COVID-19 Certificate. Luxembourg, Poland, and Ukraine utilized the official governmental app for various procedures (such as taxes and form submissions) to acquire the certificate. In Germany, citizens were required to present their vaccination card at a pharmacy to obtain a paper QR code, which could then be scanned using two apps to obtain a digital certificate.

## 4. Discussion

### 4.1. Principal Results

PHC was the first point of care in 28 countries, sometimes in collaboration with other healthcare resources. In addition, 24 countries developed apps to complement classical medical care. The most commonly implemented app was for contact tracing, followed by the Digital COVID-19 Certificate app at the time of the data assessment. It is possible that this situation changed as the pandemic progressed. Bulgaria, Italy, Serbia, North Macedonia, and Romania allowed interoperability between PHC and the COVID-19 app. Additionally, only Poland and Romania’s COVID-19 apps took into consideration social needs.

### 4.2. Comparison with Prior Work

The pandemic presented a unique opportunity for innovation in healthcare service delivery. The majority of COVID-19 cases could be managed on an outpatient basis, without requiring hospital admission [13]. In Europe, several national COVID-19 apps were developed to disseminate information to the general population and COVID-19 patients. Previous studies have focused on app characteristics such as functionality, esthetics, and information quality [14,15], as well as the profile of app users [16,17], but little attention has been given to the role of these apps in connecting patients with the healthcare system.

It is crucial to highlight that, in addition to the applications developed specifically for managing the COVID-19 pandemic, there were other telemedicine applications (tele-apps) implemented by various companies that were operating even before the pandemic [18]. Tele-apps were designed to provide continuous remote health services, and they include features such as virtual medical consultations and prescription management and will remain relevant in the long term. In contrast, COVID-19 apps, quickly developed to respond to the urgency of the pandemic, had a temporary utility, and their relevance may decrease as the health crisis is controlled. This distinction is essential to understand the impact and future sustainability of these technologies in PHC, and IT tools served as the initial contact points for citizens and COVID-19 patients. However, during phases of lockdown in some European countries, there was a slight decrease in attendance at PHC facilities [19,20]. Despite citizens using apps and hotlines to seek guidance and medical advice, several challenges arose in providing an adequate and coordinated response to COVID-19 cases. Firstly, patients may receive conflicting advice depending on the channel they use, including the hotline, COVID-19 app, or PHC professionals. Secondly, there was a risk of information repetition by citizens or COVID-19 patients, leading to varying advice provided across different contact points. This issue was exacerbated by the overwhelming workload and shortage of staff at the public health, hospital, and PHC levels [21]. Proper coordination is essential to enhance effectiveness in the initial contact with the healthcare system [22]. Thirdly, data collected in IT tools should be integrated into electronic patient health records to ensure that valuable clinical and epidemiological information provided by citizens or patients is not missed. Currently, this integration was only considered in Italy and Serbia. Therefore, the lack of connection between PHC and IT tools contributes to fragmented care [23]. Interoperability of electronic records across all these services, including health IT tools, is crucial to provide comprehensive care and ensure patient safety. In this context, the OECD recommends robust information systems that deliver high-quality care, necessitating investments and active participation from stakeholders and the community [24].

COVID-19 apps were developed in response to the health crisis, but their cost-effectiveness, especially for contact tracing, still needs to be evaluated. There is a need for more evidence to understand their role in managing contact tracing, particularly as real-time usage in Europe was lower than expected [4,8]. Additionally, since many countries conducted contact tracing through PHC or public health systems, COVID-19 apps would need to be interoperable with PHC medical records to ensure efficient contact tracing and follow-up of COVID-19 cases [1,4]. While COVID-19 apps often included features for the self-assessment of symptoms and reporting results to public health authorities, they did not always address patient-specific needs [25]. They also failed to tackle the issue of digital illiteracy, which is more prevalent in vulnerable groups, as well as undocumented individuals [6]. Requiring identity card information for app registration could serve as a barrier for marginalized groups. Among the countries studied, only Serbia allowed patients to directly ask questions to their regular GP, and Poland provided the option to connect with a social worker. These aspects are crucial in addressing patient needs and delivering effective care, especially for vulnerable populations.

The characteristics and contents of the COVID-19 apps varied across Europe [4,25]. Although these apps are regulated by the medical devices legislation in the EU [26], further measure would be necessary to ensure a minimum set of functions and integration of app information within the existing healthcare system, particularly in PHC medical records and social services records. The lack of connection between the COVID-19 app’s clinical information and the healthcare system represents a missed opportunity to provide more integrated care and facilitate referrals to PHC facilities as the primary point of care in most European countries.

The Digital COVID-19 Certificate was initially introduced to mitigate the spread of COVID-19 in international travel and other social activities. However, it is important to evaluate certain factors such as privacy concerns and social stratification [27]. One clear advantage of the Digital COVID-19 Certificate is that it promotes agreements between countries to adopt the same technology, allowing for its validity in the EU and other nations [11]. The process of obtaining the certificate has been integrated into the available IT tools in each country. It is positive that some countries enable patients to download the certificate directly through their existing patient apps, simplifying the process and allowing for additional functionalities to be added as the disease evolves. Serbia stands out as the only country to include a post-COVID-19 self-assessment questionnaire in their app, which is important in addressing the emergence of long COVID-19. However, only three countries utilized their patient record app to obtain the certificate, despite its effectiveness in expediting the response. Instead, nine countries opted to create a new app, which may inconvenience individuals by requiring them to use two different applications for the same purpose and could also incur additional costs for the countries. It would have been more advantageous if the digital certificate could have been integrated with other immunization records for international travel.

### 4.3. Implications for Research, Policy, and Practice

From a research standpoint, this study highlights the need for further investigation into the effectiveness and usability of COVID-19 apps, as well as their integration with PHC systems and impact on health outcomes. It is essential to involve PHC professionals in the creation of app protocols through advisory committees, as they can provide valuable insights into patient-centered approaches. Having practicing PHC professionals as advisors can ensure that technology integration with healthcare systems enhances patient experience, reduces costs, and maintains confidentiality and usability [28]. This should be viewed as a long-term vision, recognizing the pivotal role that PHC plays in managing and responding to health crises like the ongoing pandemic. We suggest expanding the usage of apps with integrated access to patient medical records, encompassing both primary and secondary care records. It is advised that users can access at least their current medical conditions, ongoing treatments, and vaccination statuses. Furthermore, enabling appointment requests with the involved healthcare professionals would facilitate direct communication with the health center, ensuring comprehensive patient care.

In practice, this study emphasizes the importance of seamless integration between COVID-19 apps and PHC services to enable timely and appropriate care for suspected cases. Policymakers and healthcare providers should prioritize the development of robust and interoperable app solutions that effectively connect individuals with PHC services. Having an app connected to PHC can not only facilitate patient access to PHC, but also accessing their clinical information enhances patient safety if they are treated in a different region away from their home. Having the ability to expand the functions of that app where their medical history resides would allow us to trace COVID-19 contacts, follow-up and track long COVID symptoms, and monitor possible therapies that alleviate symptoms. At the same time, it is necessary to integrate the patient’s medical history with their social history by collecting social determinants, the patient’s social circle, care needs, and interventions carried out by social services.

The experience gained from the digital certificate implementation in the EU should have been utilized to improve citizens’ access to their health data and ensure their validity and usefulness across Europe, particularly in the context of cross-border policies. Overall, the findings call for collaborative efforts among researchers, policymakers, and healthcare providers to optimize the integration of COVID-19 apps with PHC, leading to improved patient outcomes and streamlined healthcare services.

### 4.4. Strengths and Limitations

This study represents the first examination to date of how patients were able to connect with PHC services through governmental apps in Europe. The findings offer valuable insights into the opportunities and challenges associated with using apps to facilitate patient access to healthcare services. The study also provides practical examples of how these tools can be integrated into existing healthcare systems to enhance patient accessibility. The information presented in this study has been collected from publicly available and reliable online sources by local researchers who either worked in PHC or maintained close contact to GPs. Although the local researchers utilized trusted network resources to answer the questionnaires, it is possible that some relevant information may have been overlooked. In the case of the United Kingdom, the information is limited to England as no researchers were available in other regions. In Slovenia, a non-governmental app (CO-VID-Follower) was included due to its widespread usage among the population. This Slovenian app was developed by IT and healthcare professionals and incorporated information from governmental and public health sources. Although the COVID-Follower app was useful for the Slovenian population, it may have posed security risks since it was a non-governmental app. In Croatia, an app called “eCitizens” served as the patient portal for health records. Although the Croatian app allowed for information exchange between GPs and patients with mutual agreement, it was not specifically designed for COVID-19 and did not receive significant promotion or improvement during the pandemic.

The information was sourced exclusively from official channels, thereby excluding any critical aspects or malfunctions of the apps from the results. This limitation hinders a comprehensive evaluation of the apps’ value. While our primary objective was to describe the relationship between the apps and primary healthcare, the inclusion of additional functionalities would have enriched the dataset, allowing for a more comprehensive discussion of the apps’ limitations sourced from external channels. Additionally, as the data were collected retrospectively, only the latest version of each app available during the study period was considered.

## 5. Conclusions

Most European countries developed national COVID-19 apps, such as contact-tracing apps or Digital COVID-19 Certificate apps, at the onset of the pandemic. However, there was a lack of interoperability between these apps and primary healthcare information systems in most countries. This represents a missed opportunity to integrate the primary care sector, which has shouldered a significant burden in caring for patients with COVID-19, as a crucial pillar of the pandemic response. Future initiatives need to acknowledge the importance of the primary care sector and incorporate primary care into IT applications to ensure optimal and safe care for the majority of patients.

## Figures and Tables

**Figure 1 healthcare-12-01420-f001:**
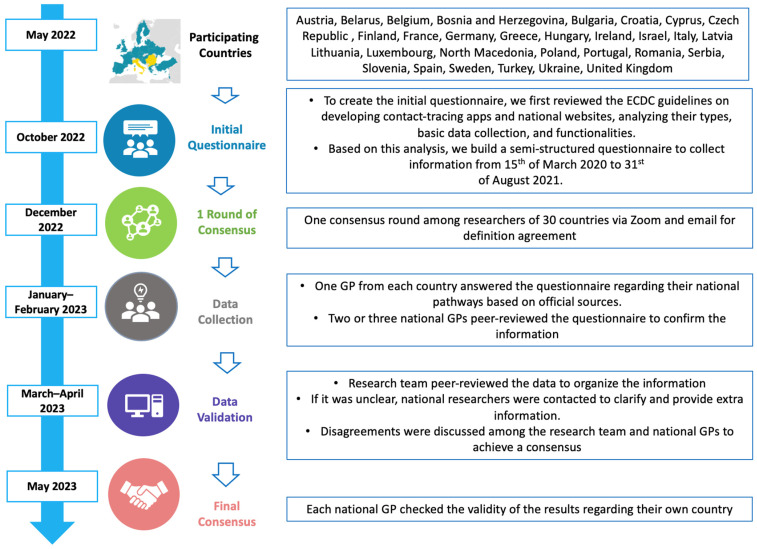
Participating countries and consensus on the questionnaire regarding the COVID-19 mobile apps and the connection with primary healthcare. Figure adapted from Ares-Blanco et al. [12].

**Figure 2 healthcare-12-01420-f002:**
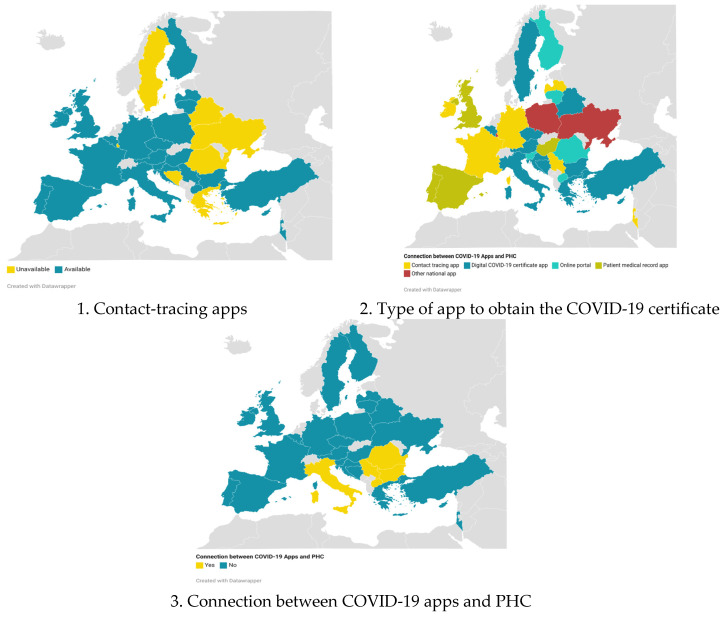
Situation of COVID-19 apps in Europe; map created with Datawrapper.

**Table 1 healthcare-12-01420-t001:** Questionnaire of the study.

Country Which was the patient’s first contact with the health system? Was there a COVID-19 hotline? Were there governmental COVID-19 apps? If there were governmental COVID-19 apps, please, answer the next questions for each app: - Name of the app? - Was the app a contact-tracing app? - Did the app have more functions than just contact tracing? If there were more functionalities, could you describe them? - Was the app connected to primary healthcare (PHC)? If the app was connected to PHC, please, explain how the connection was and which functions were offered. Were there governmental apps for the Digital COVID-19 Certificate? - If the answer is no, was there an online portal to download the certificate? - If the answer is yes, what was the name of the app? If the app was not made especially for creating the Digital COVID-19 Certificate, what type of app was used to create the Digital COVID-19 Certificate? Could you give us the references to confirm theses answers with the original sources?

**Table 2 healthcare-12-01420-t002:** Characteristics of the patient’s first contact with the health system, COVID-19 hotline, and governmental apps and connection between the apps and primary care in Europe.

Country	Patient’s First Contact with Health System	COVID-19 Hotline	Governmental COVID-19 Contact-Tracing App					Governmental App for Digital COVID-19 Certificate
			Name	Contact Tracing	Situation of the App in June 2024	Other Functions	App Connected to PHC	
Austria	GP/Hotline	Yes	Stopp Corona	Yes ^	Inactive	- Daily symptoms and health status tracker	No	Green pass app
Belarus	GP	Yes	No App	No	-	No	No	Traveling without COVID-19
Belgium	GP/A&E	Yes	Coronalert	Yes	Inactive	- Health advice related to COVID-19 - Frequently updated COVID-19 statistics	No	COVID-safe
Bosnia and Herzegovina	GP/Hotline	Yes	No App	No	-	No	No	Institute of Public Health App
Bulgaria	GP/A&E	Yes	ViruSafe	Yes	Inactive	- Health advice related to COVID-19 - Daily symptoms and health status tracker	Yes	COVID check BG
Croatia	PHC/PH/A&E/Hotline	Yes	Stop COVID-19	Yes	Inactive	- Health advice related to COVID-19 - EU Digital COVID-19 Certificate	No	CovidGO
Czech Republic	PHC	No	eRouška	Yes	Inactive	- Health advice related to COVID-19	No	čTečka
Cyprus	GP	No	CovTracer	Yes	Inactive	- Health advice related to COVID-19	No	CovPass Cyprus
Finland	PHC/Private Sector/App	Yes	Koronavilkku	Yes	Inactive	- Provided instructions on contacting healthcare personnel and other COVID-19-related instructions	No	Online portal (OmaKanta)
France	GP/Hotline	Yes	TousAntiCovid	Yes	Inactive ^Ω^	- Health advice related to COVID-19 - Frequently updated COVID-19 statistics - Reminders for COVID-19 vaccination and vaccination locations - EU Digital COVID-19 Certificate	No	TousAntiCovid
Germany	GP/Hotline	Yes	Corona-Warn-App	Yes	Active ^Ψ^	- Frequently updated incident reports for Germany or a particular region - Diary for COVID-19 symptoms - To upload your COVID-19 vaccination certificate - To generate a QR code to participate in social events	No	CovPass app * Corona-Warn-App *
Greece	PHC/Hotline	Yes	No App	No	-	No	No	COVID Free GR Wallet
Hungary	GP	No	Virus Radar	Yes	Inactive	No	No	EESZT app
Ireland	PHC, Hospital	Yes	COVID Tracker	Yes	Inactive	- Health advice related to COVID-19 - Frequently updated COVID-19 statistics and COVID-19 vaccination - Self-assessment of symptoms - EU Digital COVID Certificate	No	COVID Tracker
Israel	COVID-19 Telephone Hotline	Yes	Hamagen	Yes	Inactive	- Immunization certificate, including international certificate/digital COVID-19 Certificate	No	Hamagen app
Italy	GP/Out of Hours	Yes	Immuni	Yes	Inactive	- Health advice related to COVID-19	No	Certificazione Verde COVID-19
			e-covid Sinfonia (Campania region)	No	Active	- To send reports to the GP (suspected contagion) - To receive your COVID-19 test results - To monitor symptoms - To make an appointment for COVID-19 vaccination	Yes	
			Tests and Swabs (Campania region)	No	Inactive	- Patients can register COVID-19 test results - GPs can check COVID-19 results of patients who are under their care - Manage contact tracing - Data are shared with the Certificazione Verde COVID-19	Yes	
Latvia	GP/Hospital/112/113	Yes	Apturi COVID	Yes	Inactive	- Health advice related to COVID-19 - To monitor symptoms	No	Online portal
Lithuania	PHC/Telephone Hotline/112	Yes	KoronaStopLT	Yes	Inactive	No	No	Online portal (e. sveikata.lt)
Luxembourg	GP/Hotline/Hospital	Yes	No app	No	-	No	No	MyGuichet App (National app to any legal procedure)
North Macedonia	PHC	No	StopKorona!	Yes	NA	- Self-assessment of symptoms, health status tracker, - GPs can check the COVID-19 results and COVID-19 symptoms of patients who are under their care	Yes	Online portal (MK Wallet, Vakcinacija.mk)
Poland	PHC/Hotline/Hospital	Yes	STOP COVID—ProteGo Safe	Yes	Inactive	- Health advice related to COVID-19 - Frequently updated COVID-19 statistics - Self-assessment of symptoms	No	mObywatel (National app for any legal procedure)
			Kwarantanna domowa—quarantine	No	Inactive	- Mandatory app if you are in quarantine to describe your location. - Self-assessment of symptoms - To contact a social worker and/or sociologist in the case of need		
Portugal	PHC/Telephone Hotline/112	Yes	StayAway COVID (For patients)	Yes	Inactive	- Voluntary app for contact tracing - To inform patients about their risk exposure	No	SNS24 (app and portal) #
			TraceCOVID-19 (For HCW)	Yes	Inactive	- To introduce detailed records of specific information about cases, respective contact tracing, surveillance, and clinical follow-up of patients with suspected or confirmed COVID-19	No	
Romania	PHC/Hotline	Yes	Coronaforms	No	Active	- To introduce positive COVID-19 tests - To monitor patients with COVID-19 - To send report to the GP - To indicate the status of the patient post-COVID-19	Yes	Online portal
			Diaspora Hub	No	Active	- To help Romanians who live abroad to obtain social support and translation and share useful information during the pandemic	No	
Serbia	GP/Hotline	Yes	eHealth	Yes	Active	- Self-assessment of symptoms - To contact your GP to ask questions about COVID-19 - Post-COVID-19 questionnaire - EU Digital COVID-19 Certificate	Yes	Online portal
Slovenia	PHC/Hotline/PH	Yes	Case and vaccination registration app	No	Inactive	- Data were sent to an online portal to download the COVID-19 Certificate	No ##	Online portal
			COVID-Follower **	No	Inactive	- Health advice related to COVID-19 including vaccination - Frequently updated COVID-19 statistics - Self-assessment of symptoms	No	
			OstaniZdrav	Yes	Inactive	- Health advice related to COVID-19	No	
Spain	GP/A&E	Yes	Radar COVID	Yes	Inactive	- Health advice related to COVID-19	No	Regional patient’s apps (e-medical history, e-prescription, etc.) or online portal
Sweden	PHC/Hotline	No	No app	No	-	No	No	Covidbevis/COVID certificate
Turkey	PHC, Hotline	Yes	Hayat Eve Sığar (HES)	Yes	Active	- Health advice related to COVID-19 and self-assessment of symptoms - Frequently updated COVID-19 statistics and risk density on the map - Notification to the relevant authorities when leaving isolation - Check a relative’s health status by querying his/her code - Healthcare settings locations - To obtain a QR code to enter in shops/social events	No	HealthPass (COVID-19 vaccination, tests, and immunization certificates)
Ukraine	GP	Yes	No app	No	-	No	No	Diya app (national app for any legal procedure)
United Kingdom	Phone line/online platform		NHS COVID-19	Yes	Inactive	- Health advice related to COVID-19 - Self-assessment of symptoms - General information on COVID-19	No	NHS app

A&E: Accident & Emergency department; Digital COVID-19 Certificate or EU Digital COVID-19 Certificate: digital proof that a person has either been vaccinated against COVID-19 or received a negative test result or recovered from COVID-19; EU: European Union; GP: General Practitioner; HCWs: Healthcare workers; NA: information not available; PH: public health; Online portal: Online patient platform; ^ The app was used to contact trace on an individual level. The information was not officially collected by health authorities for contact tracing; * In Germany, to obtain an EU Digital COVID-19 Certificate, you first have to go to a pharmacy with your COVID-19 vaccination card in order to obtain a vaccination certificate with a QR code on paper. Once you have the QR code, you can use the CovPass App or the Corona-Warn-App to have a digital certificate issued; ** It is not a governmental app, but the use was generalized in Slovenia; # SNS24 is a generic national app covering a range of services, including vaccine card, sick leave, prescriptions, medical certificate of multipurpose disability, pathologies (allergies and rare diseases), exams, clinical referrals, usual medication consultation, usual medication ordering, contact your health facility, contact with the SNS 24, teleconsultation (through RSE Live), and EU Digital COVID-19 Certificate; ## COVID-19 testing and COVID-19 vaccination transferred information to the electronic health record (EHR), but there was no connection between clinical data from the apps and the HER; Ω: The app is currently on hold with reactivation on demand according to the COVID-19 epidemiological status; ^Ψ^: The app is technically functional, but the statistics have not been updated since 1 June 2023.

## Data Availability

Any data produced or examined during this study are available for sharing upon request. The primary data can be directly reviewed in the Appendix A files. All methodologies were conducted in strict adherence to pertinent guidelines and regulations. For additional inquiries, please contact the corresponding author.

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
