# Peer review of "The Use of COVID-19 Mobile Apps in Connecting Patients with Primary Healthcare in 30 Countries: Eurodata Study"

_healthcare, 2024, doi:10.3390/healthcare12141420_

Round 1

Reviewer 1 Report

Comments and Suggestions for Authors

Regarding the manuscript with the title  "The Use of COVID-19 Mobile Apps in Connecting Patients with Primary Health Care in 30 countries: Eurodata study."

I would like to inform the authors that this manuscript is complete. It has an acceptable methodology. The collected material is very good and the organization of the material is appropriate. But relevant references are needed to complete this manuscript. For example, the following reference can be used in this study. This reference specifically deals with mobile phone applications regarding tracing patients and communicating with patients with or suspected of covid-19. 

"Mobile Apps for COVID-19 Detection and Diagnosis for Future Pandemic Control: Multidimensional Systematic Review"

Other than this, it seems essential that this manuscript is very well written and complete in every way.

Author Response

Thanks to the reviewer for the perspective, we have included the recommended reference. Thank you for the suggestion.

Reviewer 2 Report

Comments and Suggestions for Authors

The authors present The Use of COVID-19 Mobile Apps in Connecting Patients with Primary Health Care in 30 countries. The objective of the study stated in the abstract is “To describe if governmental COVID-19 apps allowed COVID-19 patients to connect with primary healthcare (PHC) through their apps in Europe and to examine how the digital COVID-19 certificate was obtained”. Details sections are presented to cover different parts. Some points need to be addressed to clarify the objective and the achievements of the study.

The above-stated sole objective of the study does not add any significant knowledge to the research community.

The two main parts of the objective are the connectivity of the app with PHC and digital certificates. The findings can be more useful if the details of the information obtained in the PHC are used for other healthcare applications.

The tracing, social distance, and other parameters of COVID-19 may not be relevant to other healthcare-related issues. How the use of this information is useful for other healthcare units. There is a need to correlate the findings with other healthcare-related issues to make a useful impact.

In lines 334-335, the authors state that the integration with e-health records was done in only two countries. The reasons for not integration by the other countries need to be highlighted e.g., privacy concerns that may be useful in scenarios where such apps may be needed.

The study was conducted around four years ago. Are these apps being used presently? What is the frequency of using the apps? What are the main purposes of the apps? How the findings of the study can be used in other scenarios related to e-health or pandemics that may be different from COVID-19?

Author Response

Thank for your comment and for the time reviewing our manuscript. We understand your perspective, as GPs we are in an excellent position to see the COVID-19 patients’ pathway through the healthcare system and the blind spots in the healthcare system to address their needs. We have already published the clinical pathway in 30 countries, the type of COVID-19 indicators in primary care in 31 countries and we have more papers under review at the moment. One of our key findings is that the care provided across Europe was very heterogenous and the burden of care in PHC has not been described properly in most of the countries. This is crucial as most of COVID-19 patients were diagnosed and treated in the community and in case they were admitted, they were referred back to the community. That happened in a moment of scarcity of healthcare professionals. At that moment, the possibility to use Apps to decrease the workload in the health system was seeing as a useful tool. The countries spent special funding to support these apps. The Coronavirus Response Investment Initiative gives Member States an upfront cash injection of

EUR 8 billion from the EU cohesion funds which could accelerate up to EUR 37 billion of European public investment to fight the coronavirus.

As the patients were in the community, we found relevant to describe if this COVID-19 apps were helping patients to connect with primary care and if they could be used as a tool to improve the clinical care of the patients. We found the COVID-19 apps did not help much in this function. We don’t support to have several apps for health purposes. In our opinion, patients should have only one app with all the functionalities included where they could have their COVID-19 tests, their clinical course, health promotion advices, vaccination record, treatments records and at least a way to get appointments with primary care. We believe that the access to health apps should be as simple as possible to avoid difficulties to population with less digital capacities.

GPs are focused in the whole person, this perspective has proved to increase the survival of the population, decrease the use of the emergency department and the hospitalization rate. As clinicians who provide clinical care, we think contact tracing, social distance and other COVID-19 parameters are relevant to provide a comprehensive care.  It is important to provide health promotion advice related to COVID-19, if the advice was given correctly, the doctor/nurse at other sections of the healthcare system (the emergency hotline, at the Emergency room or at the internal medicine ward) don’t have to repeat it again. Also, having access to the symptoms or clinical course of the patient can save time if the patient is sent to the Emergency room. We agree that there is a need to correlate the findings with other healthcare-related issues to improve the care and that is one of the reasons, the design of the apps have to be improved to be able to study this correlation and possible benefit.

Most of the apps are not used anymore, that is also an important result as many of the apps were only designed for the COVID-19 pandemic they lost the possibility to be used with other functions in the long term. These apps were download for millions of European citizens and their extensive impact could have been higher if countries would at least have considered other functionalities as a general vaccination record and connected them to general practitioners of PHC facilities. In an European assessment report, no reference was made to the possibility of linking those app with PHC system although some apps were linked among countries through a European platform There are other studies that had described the use of COVID-19 apps and it was not our aim to repeat the same research. These results are important for future pandemic and other health conditions, we have learnt:

  • It is relevant provide interoperability between PHC records and the apps.
  • It is important to facilitate patients to contact with their primary care provider through the apps.
  • It is important to consider the social needs of the patients in the apps to provide a comprehensive care.

 References:

  1. Goverment of Ireland. Coronavirus response investment initiative. Available at: https://eufunds.ie/covid/coronavirus-response-investment-initiative/
  2. Sandvik H, Hetlevik Ø, Blinkenberg J, Hunskaar S. Continuity in general practice as predictor of mortality, acute hospitalisation, and use of out-of-hours care: a registry-based observational study in Norway. Br J Gen Pract. 2022 Jan 27;72(715):e84-e90. doi: 10.3399/BJGP.2021.0340. PMID: 34607797; PMCID: PMC8510690.
  3. Alexandra Prodan, Strahil Birov, Viktor von Wyl, Wolfgang Ebbers. Digital Contact Tracing Study. Study on lessons learned, best practices and epidemiological impact of the common European approach on digital contact tracing to combat and exit the COVID-19 pandemic VIGIE 2021-0649 Framework Contract SMART 2019/0024, Lot 2. European Commission. 2022. Doi: doi: 10.2759/146050. [cited 28.06.2024]. Available at:

Reviewer 3 Report

Comments and Suggestions for Authors

Thank you for the opportunity to review the manuscript. Congratulations! This subject is current and relevant to be applied in future pandemics. I would like to leave my suggestions to the authors.

 -Line 168: Please describe “GP” (general practitioner).

 -Lines 182-183: “The questionnaire also collected information on the use of information technology (IT) tools to support medical care (Supplement 3).”  The authors could write this information in the article and not as supplement.

 -I did not find “supplement 5” in the text.

 -Discussion: Please write a brief summary of the study and results in the first paragraph.

Author Response

Thank for your kind words. We have written properly General Practitioner in the text. We have included the supplement 3 as a table 1 in the test. We are sorry we missed quoting the supplement 5 in the text, we have included at the end of the methods section as supplement 4 as the questionnaire is now a table in the text and not a supplement. We wrote a brief summary of the study in the first paragraph of the results:

“PHC was the first point of care in 28 countries, sometimes in collaboration with other healthcare resources. In addition, 24 countries developed apps to complement classical medical care. The most commonly implemented app was for contact tracing, followed by the Digital COVID-19 Certificate app at the time of the data assessment. It is possible that this situation changed as the pandemic progressed. Bulgaria, Italy, Serbia, North Macedonia and Romania allowed interoperability between PHC and the COVID-19 app. Additionally, only Poland and Romania's COVID-19 apps took into consideration social needs”

Reviewer 4 Report

Comments and Suggestions for Authors

Thank you for inviting me to review this manuscript. I congratulate the authors on their work and I have some comments:

  1. The abstract mentions PHC twice as primary healthcare. Please review this carefully and check the abbreviations throughout the manuscript. Abbreviations should be defined the first time they are mentioned. For example, on lines 416 and 309, please ensure consistency in the use of abbreviations. This also applies to IT tools.
  2. As I understand it, these were governmental applications, correct? Does each country have its own application?
  3. Please include a diagram detailing how the data was collected, i.e., the steps you performed, so that the process can be easily reproduced.
  4. The abbreviations in Table 1 need clarification. Please format them as follows: Abbreviations: EU - European Union, GP - General Practice, HCW - Healthcare Workers, PH - Public Health, Online Portal - Online Patient’s Platform.
  5. It would be beneficial to add a column in the table (Active Status) to indicate whether the application is currently active as of July 2024, for example. Alternatively, this information could be included in the discussion if all applications remain active. This is particularly important for countries like Slovenia that have private apps, as maintaining these apps requires ongoing payment. This will necessitate a discussion on the role of governmental funding for such projects in the future and its timeframe.
  6. Please move the Strengths and Limitations section to the end of the manuscript, before the Conclusion.
  7. In the discussion, please highlight that other tele-apps/websites were also developed by companies (https://www.mdpi.com/2227-9032/10/2/385). Discuss the differences between tele-apps and COVID apps. As you may know, tele-apps gained a lot of interest during COVID and they were operational even before that. The issue with COVID apps is that they may only serve for a limited time.
  8. In the limitations section, it’s good that you mentioned the integration of COVID apps with PHC services. I believe some countries have everything linked to a single governmental portal using the user’s social security number (or similar). However, there would be a security risk if the application was developed on a non-governmental portal. This point could also be added. In short, discuss why it is a priority for governments to develop such apps.

Comments on the Quality of English Language

English is fine; only multiple issues with abbreviations 

Author Response

 Thank you for taking the time to read our study and for your comments.

  1. We have reviewed all the PHC abbreviatures and we wrote properly “the primary health care”, thanks for noticing and telling us. We have also homogenized the term “apps” through the text.
  2. The apps were governmental apps with the exception of Slovenia that it was a private app for public use. Each country developed their own app under their own criteria. An official list of Apps is available in the report: European Commission. Digital Contact Tracing Study. Study on lessons learned, best practices and epidemiological impact of the common European approach on digital contact tracing to combat and exit the COVID-19 pandemic VIGIE 2021-0649 Framework Contract SMART 2019/0024, Lot 2 . Available at: https://commission.europa.eu/system/files/2023-02/DigitalContactTracingStudy.pdf
  3. We have included a diagram with the data collection.
  4. The abbreviations have been modified following your advice.
  5. We have included a column in the table to describe if the contact tracing app is active or not. As the certificates are not needed anymore, we have not included this information and we have quoted in the results
  6. We moved the Strengths and Limitations to the end of the section.
  7. We have added the proposal to the text as: “it is crucial to highlight that, in addition to the applications developed specifically for managing the COVID-19 pandemic, there were other telemedicine applications (tele-apps) implemented by various companies and operating even before the pandemic, (REF https://www.mdpi.com/2227-9032/10/2/385). Tele-apps, designed to provide continuous remote health services, include features such as virtual medical consultations and prescription management, and will remain relevant in the long term. In contrast, COVID apps, quickly developed to respond to the urgency of the pandemic, have a temporary utility and their relevance may decrease as the health crisis is controlled. This distinction is essential to understand the impact and future sustainability of these technologies in healthcare.”
  8. We have included your suggestion in the limitations as “Although the COVID-Follower app was useful for the Slovenian population, it may have posed security risks since it was a non-governmental app.”

Round 2

Reviewer 2 Report

Comments and Suggestions for Authors

Majority of the comments are addressed in the revised version.